# RDHNet: Addressing Rotational and Permutational Symmetries in Continuous Multi-Agent Systems

## Abstract

Symmetry is prevalent in multi-agent systems. The presence of symmetry, coupled with the misuse of absolute coordinate systems, often leads to a large amount of redundant representation space, significantly increasing the search space for learning policies and reducing learning efficiency. Effectively utilizing symmetry and extracting symmetry-invariant representations can significantly enhance multi-agent systems' learning efficiency and overall performance by compressing the model's hypothesis space and improving sample efficiency. The issue of rotational symmetry in multi-agent reinforcement learning has received little attention in previous research and is the primary focus of this paper. To address this issue, we propose a rotation-invariant network architecture for continuous action space tasks. This architecture utilizes relative coordinates between agents, eliminating dependence on absolute coordinate systems, and employs a hypernetwork to enhance the model's fitting capability, enabling it to model MDPs with more complex dynamics. It can be used for both predicting actions and evaluating action values/utilities. In benchmark tasks, experimental results validate the impact of rotational symmetry on multi-agent decision systems and demonstrate the effectiveness of our method.

## 1 Introduction

In recent years, multi-agent reinforcement learning (MARL) has made significant advancements and has been widely applied in areas such as traffic planning (Mushtaq et al. (2023); Ma et al. (2024)), power management Keren et al. (2024), gaming (Zhai et al. (2021); Li et al. (2018); Berner et al. (2019)), and Robotics (Wen et al. (2022); Chen et al. (2023)). However, compared to single-agent reinforcement learning, MARL faces more challenging issues, including credit assignment( Yarahmadi et al. (2023; 2024)), scalability( Ying et al. (2024); Ma et al. (2024)), and imperfect information games( Zhang et al. (2024)). In this paper, we primarily focus on the symmetry problem in MARL.

Symmetry is widely present in the natural world, and many studies in fields such as image processing, graph neural networks, and point clouds have focused on symmetry. In multi-agent reinforcement learning (MARL), research on symmetry is still in its infancy. Symmetry can be broadly categorized into permutation symmetry and rotational and mirror symmetry (in this paper, we refer to both rotational and mirror symmetry as rotational symmetry, as they can be derived from orthogonal transformations). Some research has been conducted on permutation symmetry in MARL (Wang et al. (2019); Hu et al. (2021); Jianye et al. (2022); Zhou et al. (2022)), and HPN (Jianye et al. (2022)), with the most notable work being HPN. By designing a network structure with permutation-invariant inductive bias, permutation-invariant representations can be naturally obtained, leading to the corresponding action outputs. However, research on rotation invariance (RI) in MARL is still in its early stages.

Rotational symmetry is prevalent in the real world. For instance, if a person learns traffic rules at a driving school, they can easily generalize them to real-world traffic conditions without considering whether they are facing north or south. Similarly, a basketball team, after training, can play in a new arena without worrying about the actual orientation of the court. They only need to know their relative positions to teammates, opponents, and the basket. Humans naturally incorporate prior

knowledge of rotational symmetry into specific tasks, resulting in higher sample efficiency when learning a new skill. However, this is not the case for machines. Without special consideration of rotation invariance, machines cannot generalize knowledge learned from state $s$ to a rotated state $s'$, shown in Fig. 1.

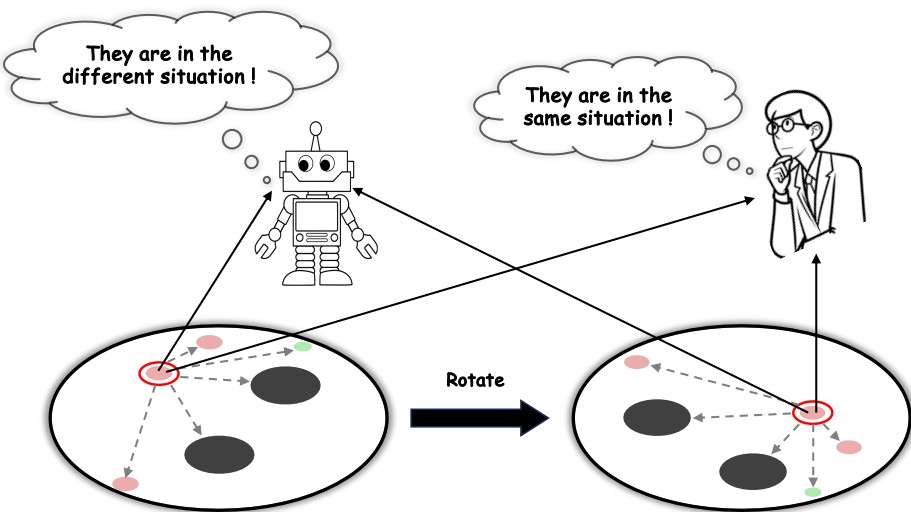

Figure 1: This figure illustrates the issue of redundant representations caused by rotational symmetry. In the figure, nodes of different colors and sizes represent various entities. At a particular moment, state $s$ is rotated by a random angle, resulting in state $s'$. If the human is the agent marked by the red circle in the figure, they can easily perceive the equivalence between $s$ and $s'$ and take appropriate actions. However, suppose the agent marked in the figure is a robot. In that case, it may fail to recognize the relationship between $s$ and $s'$, making it difficult to generalize the knowledge gained from one state to another, demonstrating how rotation can create redundant state representations in MARL.

Some works( van der Pol et al. (2021); Yu et al. (2024; 2023)) define the symmetry method and quantity for the agent's coordinate system, which improves the algorithm's performance on specific MARL tasks. However, it still cannot handle continuous rigid transformations in Euclidean space. Continuous rigid transformations involve rotating the agent's coordinates around a fixed point by arbitrary angles rather than assuming rotations are only multiples of 90 degrees or simple mirror reflections. Continuous transformations are more representative of real-world scenarios. Theoretically, if a reasonable model or algorithm could account for rotational symmetry, an agent could generalize knowledge learned from state $s$ to the rotated state $s'$, thereby significantly reducing the state representation space.

We propose a *Relative Direction Hypernetwork* (RDHNet) architecture to extract relative directional and positional information and then use a symmetry computation module to aggregate this information. Effectively leveraging symmetry compresses redundant representation space, facilitating better knowledge sharing among agents and enhancing learning efficiency and robustness. Additionally, we utilize the high-order modeling capabilities of the hypernetwork module to improve the model's expressiveness, enabling it to handle tasks with more complex dynamics. RDHNet can be used to construct not only an action value/utility evaluation network but also a policy network that produces actions based on input states.

We conducted experiments on two continuous action tasks, *Cooperative Prey Predator* and *Cooperative Navigation*. The results demonstrate that, compared to other methods, RDHNet achieves superior performance. Moreover, during execution, RDHNet does not require absolute coordinate and directional information; it can make appropriate decisions based solely on relative observations from the agent's perspective, indicating that in situations where geographic positioning information is unavailable, our method remains effective, whereas other methods may fail to function. We summarize our contributions as follows:

- We formalized the symmetry problem in multi-agent systems, classifying it into permutation symmetry and rotational symmetry, providing a clearer problem definition for future research on symmetry.

- We propose a network architecture that can eliminate the redundant representation space caused by symmetry without significantly reducing the network's expressiveness.

- We compare different methods on benchmark tasks to analyze and confirm the real impact of symmetry on MARL problems. We also conduct separate ablation studies for rotation invariance (RI) and permutation invariance (PI).

## 2 RELATED WORKS

### 2.1 SYMMTRY IN GNNs AND POINT CLOUDS

Recent advances in symmetry studies within Graph Neural Networks (GNNs) and point cloud processing have significantly influenced the development of invariant models to permutations, rotations, and translations. For instance, the introduction of Graph Convolutional Networks (GCNs) (Kipf & Welling (2016)) laid the foundation for permutation invariance in GNNs. In the context of point clouds, PointNet (Qi et al. (2017a)) and its successor PointNet++ (Qi et al. (2017b)) pioneered approaches that handle unordered point sets directly, enabling permutation invariance.

Building on these foundational works, recent studies have explored rotational and translational equivariance in graph neural networks (GNNs) and point cloud networks. Tensor Field Networks (Thomas et al. (2018)) and SE(3)-Transformers (Fuchs et al. (2020)) exemplify this trend by incorporating roto-translation equivariant features, enabling more accurate and robust processing of 3D point clouds and molecular structures. Additionally, work (Maron et al. (2019)) extended the expressive power of GNNs by proving the capability of certain architectures to universally approximate permutation-invariant functions, further advancing the theoretical understanding of symmetry in GNNs. DimeNet (Gasteiger et al. (2020)) captures angular dependencies between atoms in a molecular graph, which enhances the network's ability to model geometric structures without being affected by the specific orientation or position of the molecule in space.

### 2.2 SYMMTRY IN MARL

Some works have begun to address the issue of redundant space caused by symmetry in multi-agent reinforcement learning (MARL). Methods such as ASN (Wang et al. (2019)), UPDeT (Hu et al. (2021)), and HPN (Jianye et al. (2022)) have focused on the redundancy problem due to permutation symmetry. They modify the actor network by incorporating prior knowledge to compress the redundant representation space caused by permutation order. The primary differences among them are: ASN (Wang et al. (2019)) uses MLP as the basic network module, UPDeT (Hu et al. (2021)) uses transformers, and HPN (Jianye et al. (2022)) uses hypernetworks. Essentially, they all integrate permutation symmetry into the network inductive bias to address the redundancy problem caused by different permutations of semantically identical states. However, these approaches do not address the issue of rotational symmetry.

Work(van der Pol et al. (2021)) was the first to address the rotational symmetry issue in MARL, which involves symmetry operator pairs $(L_g, K_g)$ to the model. Works (Yu et al. (2023; 2024)) generate rotated samples during training and introduce a rotational symmetry loss function to exploit symmetry. However, a major limitation of these works is that they can only handle rotational symmetry at multiples of 90 degrees. They neither consider nor can be applied to continuous random rotational symmetry, which is precisely the focus of our work and is more aligned with real-world scenarios.

## 3 PROBLEM FORMULATION

A Multi-Agent Reinforcement Learning (MARL) problem is often considered a Markov decision process, which is a tuple of the form $(\mathcal{N}, \mathcal{S}, \mathcal{O}, \mathcal{A}, T, R, \rho, \gamma, \pi)$. Here, $\mathcal{N}$ is the set of all agents, $\mathcal{S}$ is the set of states, $\mathcal{O} = \mathcal{O}^1 \times \mathcal{O}^2 \ldots \times \mathcal{O}^{|\mathcal{N}|}$ is the set of joint observations where $\mathcal{O}^k$ indicates the

set of $k$-th agent' observations, $\mathcal{A} = \mathcal{A}^1 \times \mathcal{A}^k \times \ldots \times \mathcal{A}^{|\mathcal{N}|}$, $T : \mathcal{S} \times \mathcal{A} \times \mathcal{S} \to [0, 1]$ is a transition function that specifies the probability of reaching state $s' \in \mathcal{S}$ after all agents taking their joint action $\boldsymbol{a}$ in state $s$, reward function $R : \mathcal{S} \times \mathcal{A} \to \mathbb{R}$ is to return a reward value to agents to evaluate the quality of the agent's action, $\rho$ is the initial state distribution, $\gamma$ is the discount factor. At one timestep, each agent needs to take action to the environment based on their observation, and then the state of the environment will change naturally. After this, agents will be rewarded $r = R(s, a)$ and new observations. A joint policy $\pi$ is a function that maps joint observation $\boldsymbol{o} \in \mathcal{O}$ to joint action $\boldsymbol{a} \in \mathcal{A}$. $\pi$ often can be divided into a set of all agents' policies $(\pi^1, \ldots \pi^k \ldots, \pi^{|\mathcal{N}|})$, where $\pi^k$ is the policy function that maps $k$-th agent' observation to its action. The core objective of MARL is to find the optimal policy $\pi^* = \arg\max_\pi \sum_{t=0}^{\infty} \mathbb{E}_{(o_t, a_t) \sim \rho_\pi} [\gamma^t r(s_t, a_t)]$, where the subscript $t$ represents the variable value at the $t$-th timestep. However, if we directly handle a MARL problem as formalized above without addressing symmetry, it typically results in a substantial amount of redundant representations, leading to low sample efficiency.

**Permutation Symmetry** occurs when the order of agent information does not affect the overall observation information, resulting in redundant representation space due to the fixed order setting. Let $o_i = \boldsymbol{e} = [e_1, e_2, \ldots, e_m]^T$, where($e_k$ denotes the information of the $k$-th entity (an entity can be an enemy, a teammate, or other things that can influence agent making decision) observed by agent $i$. Specifically, let $\boldsymbol{e} = [e_1, e_2, \ldots, e_m]^T = \boldsymbol{x} = [x_1, x_2, \ldots, x_m]^T$. When the order of the entities' information in $o_i$ changes, i.e., $\boldsymbol{e} = [e_1, e_2, \ldots, e_m]^T = [x_1, x_2, \ldots, x_m]^T \cdot g, g \in G$, where $G$ is the set of all permutation matrices and $g$ is a specific permutation matrix in $G$. It can be seen that regardless of the value of $g$, the physical information contained in $o_i$ remains unaffected. In other words, the permutation order of the entity information is actually redundant in the decision-making process of agent $i$. However, if we do not address this issue, the permutation-induced symmetry will lead to a large amount of redundant representation space, i.e., $\boldsymbol{x} \cdot g_1, \boldsymbol{x} \cdot g_2, \ldots$ will all be recognized by the model as different observation information, thus resulting in redundant space. A function $f : X \to Y$, where $X = [x_1, x_2, \ldots, x_m]^T$ and $Y = [y_1, y_2, \ldots, y_m]^T$. If $f$ satisfies $f([x_1, x_2, \ldots, x_m]^T * g) = f([x_1, x_2, \ldots, x_m]^T) = [y_1, y_2, \ldots, y_m]^T, \forall g \in G$, then we call $f$ a permutation-invariant function. If we can construct a model that satisfies the properties of the $f$ function, then permutation symmetry can be effectively addressed.

**Rotational Symmetry** refers to the condition where rotating a state as a whole does not affect the relative physical information contained within that state. This implies that many rotational mappings of the same state lead to representational space redundancy. Let us further refine the observation $o_i$. We distinguish the features of each entity into spatial coordinates $p_k$ and other information $z_k$, so the observation can be represented as $o_i = [[p_1, z_1]^T, [p_2, z_2]^T, \ldots, [p_m, z_m]^T]$. We then reconstruct $o_i$ as $[[p_1, p_2, \ldots, p_m]^T, [z_1, z_2, \ldots, z_m]^T]$. Consider a binary function $h : (P, Z) \to Y$, where $P = [p_1, p_2, \ldots, p_m]^T, Z = [z_1, z_2, \ldots, z_m]^T$, and $Y = [y_1, y_2, \ldots, y_m]^T$. $U$ is the set of all $m \times m$ rotation matrices, and $u \in U$ is a specific rotation matrix. If $h(P \cdot u, Z) = h(P, Z) = Y, \forall u \in U$, then $h$ is called a rotation-invariant function. If a model satisfies the properties of the $h$ function, we can say that the model possesses rotational invariance.

# 4 ROTATIONAL-INVARIANT MULTI-AGENT REINFORCEMENT LEARNING

Rotation-invariant multi-agent reinforcement learning (MARL) does not require an absolute coordinate reference system; it only considers the relative positional relationships between entities. This approach aligns more closely with human intuition and facilitates better knowledge generalization. This chapter provides a detailed overview of the RDHNet framework's structure.

## 4.1 DIRECTIONAL INFORMATION PROCCESS

To address the issue of rotational symmetry, it is first necessary to distinguish which elements are direction-related and which are direction-independent in a decision-making scenario involving multiple entities. Information such as the volume and mass of entities is typically direction-invariant. In contrast, the position, motion direction, and force direction of entities are closely related to the reference direction (changing with the reference frame).

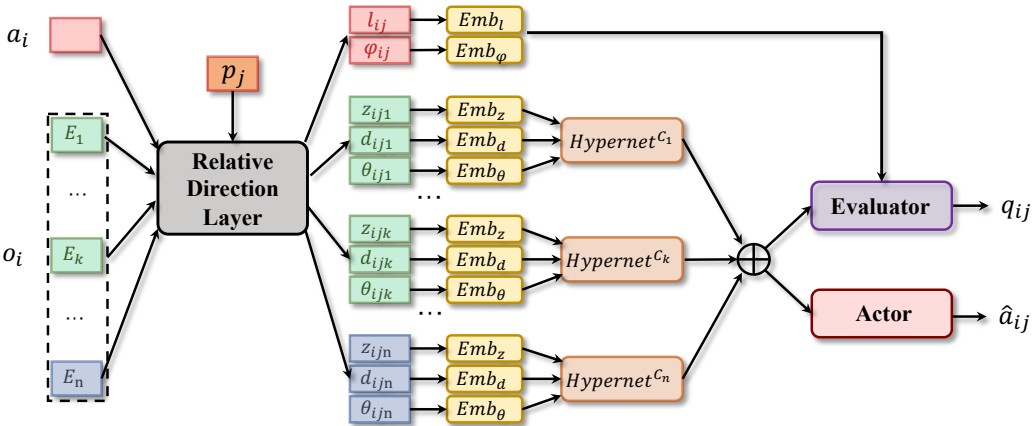

Figure 2: The overall framework of RDHNet. Initially, the information $E_k$ of the $k$-th entity in the agent's observation is decomposed into coordinates $p_k$ and direction-independent information $z_k$. Then Relative Direction Layer (RDL) maps the positions of entities $i$ and $j$, converting the Cartesian coordinates $P_k$ into the corresponding polar coordinates $(d_{ijk}, \theta_{ijk})$. If we need RDHNet to output the action value/utility, the agent's action vector $a_i$ should also be transferred to $l_{ij}$ and the relative angle $\phi_{ij}$ by the RDL. Each type of information is then encoded by its respective encoder and input into the hypernetworks for each entity type. The symmetry module aggregates this information to obtain direction-invariant representations. Finally, these representations are input into the actor or critic module to output the corresponding actions $a_{ij}$ or action values $q_i$.

Fig. 3 shows the observation of agent $i$ at a specific moment in the *Cooperative Prey Predator* task. In this observation, it can be seen that regardless of the choice of reference direction, the angles and lengths of the lines connecting each entity to agent $i$ do not change. We need to leverage this invariance in multi-agent scenarios to compress the representation dimensions of observation information. This involves deconstructing and reorganizing the original information structure, replacing the reference frame based on absolute direction with a reference frame defined by the line connecting a specific entity and the agent itself.

As an example, consider the observation shown in Fig. 3. First, a reference entity $j$ is selected. Then, with agent $i$ as the origin and the line between agent $i$ and entity $j$ as the zero-degree direction, a polar coordinate system is established, and the polar coordinates of each entity in this system are calculated. After the Relative Direction Layer (RDL), the information of $k$-th entity $E_k$ should be processed into a tuple $(z_{ijk}, d_{ijk}, \theta_{ijk})$ by the following equation:

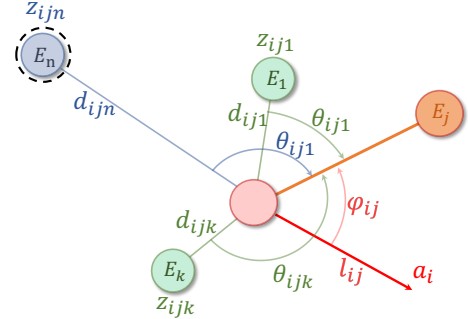

Figure 3: An observation from agent in *Cooperative Prey Predator*. This figure is an example of an observation, where circles of different colors represent different types of entities (corresponding to Fig. 2). Regardless of the state's rotation, the variables shown in the figure remain unchanged in the polar coordinate system where the zero axis is defined by the agent and entity $j$. Specifically, this means that the length of the line segments connecting each entity to the agent and their angles with respect to the zero axis are invariant.

$$z_{ijk} = z_k, \quad d_{ijk} = (x_k - x_i, y_k - y_i), \quad \theta_{ijk} = \arctan\frac{y_k - y_i}{x_k - x_i} - \arctan\frac{y_j - y_i}{x_j - x_i}, \quad (1)$$

where $z_{ijk}$ represents some attributes of entity $k$ that are independent of directional changes, while $d_{ijk}$ and $\theta_{ijk}$ correspond to the length and angle, respectively, in the polar coordinate system of entity $k$. In this system, the zero axis is defined by the line from entity $i$ to entity $j$, with the

pole located at the position of entity $i$. In practical processing, since $arctan(x)$ is an odd function with a period of $\pi$, it results in two possible values within a $2\pi$ range. This requires discarding one of the values based on the specific context, complicating efficient algorithm execution. The same issue arises when $arcsin(x)$ and $arccos(x)$ are used independently. To address this issue, we propose replacing $\theta_{ijk}$ in Equation (1) with $m_{ijk} = [\sin(\theta_{ijk}), \cos(\theta_{ijk})]$ for representing the angle information, for the detailed mathematical derivations, please refer to the Appendix A.

If subsequent evaluation of action is needed, the actions must also be processed in the "Relative Direction Layer". Here, the action (usually a force vector) of agent $i$ originally based on the absolute coordinate system is converted into a tuple $(l_{ij}, \phi_{ij})$ based on the aforementioned polar coordinate system.

At this point, the information of each entity has been separated and processed individually. After passing through the Relative Direction Layer, the positional information of entities in the environment is fully decoupled from the absolute coordinate system, making their location data solely dependent on their relative positions to other entities.

## 4.2 INFORMATION EMBEDDING AND AGGREGATION

Once the direction-dependent and direction-independent information of each entity is separated, the next step is to encode and aggregate this entity information. As shown in Fig. 2, when using entity $j$ as a reference, an information tuple $(z_{ijk}, d_{ijk}, m_{ijk})$ for entity $k$ can be obtained. Each element of the tuple is encoded in different ways: the direction-independent information $z_{ijk}$ is encoded using a multilayer perceptron (MLP), denoted as $Emb_z$; the distance information $d_{ijk}$ is encoded using radial basis functions (RBF), denoted as $Emb_d$; and the angle information is encoded using sine and cosine functions. For action value evaluation, the action vector $a_i$ is similarly processed, with the relative angle $\phi_{ij}$ encoded using sine and cosine functions and the magnitude of $a_i$ encoded using RBF. Here, every entity's information is encoded into $e_{ijk} = [Emb_z(z_{ijk}), Emb_d(d_{ijk}), Emb_\theta(m_{ijk})]$. The encoded information $e_{ijk}$ of entity $k$ inputs into a specific hypernetwork to obtain an overall abstract representation of the entity. To enhance the network's expressive capability, different hypernetworks are used for different entity categories. For example, if the category of $k$ is $C_k$, the corresponding hypernetwork $Hypernet^{C_k}$ is used, whose structure is generally consistent with that in ( Jianye et al. (2022)). Finally, the abstract representations of each entity are aggregated using a symmetric aggregation function, which ensures that the output is invariant to the permutation of input variables (i.e., a permutation-invariant function $f$ as defined in section. 3). This representation can be computed as:

$$\tilde{o}_{ij} = \sum_{k \in \mathcal{N}(i), k \neq j} \mathcal{M}_H^{C_k}\big(\text{Emb}_z(z_{ijk}), \text{Emb}_d(d_{ijk}), \text{Emb}_\theta(m_{ijk})\big), \tag{2}$$

where $\mathcal{M}_H^{C_k}$ refers to the hypernet for the class of entity $k$, and $\mathcal{N}(i)$ is an entity set that contains all the entities in the entity $i$'s view. By this point, we have effectively utilized both permutation symmetry and rotational symmetry, achieving compression of redundant representations $\tilde{o}_{ij}$ for entity $i$'s observation based on the position of entity $j$ (i.e., the symmetric physical information is always processed to yield the same representation). Here, if it is necessary to evaluate the action utility value, $a_{ij}$ will be similarly calculated.

## 4.3 ALGORITHM INPLEMENTATION

Theoretically, RDHNet can be combined with any mainstream continuous-action MARL algorithm. However, in practice, we chose COMIX (Rashid et al. (2020)), which currently demonstrates state-of-the-art performance in continuous multi-agent tasks, as the foundation for our implementation. We adopt the COMIX scheme in FACMAC (Peng et al. (2021)), which employs the cross-entropy method (CEM) (De Boer et al. (2005)) by QMIX-style (Rashid et al. (2020)). Specifically, both observations and actions are processed to obtain their corresponding rotation-invariant representations $\tilde{o}_{ij}$ and $\tilde{a}_{ij}$. Action utility value is computed by $q_{ij} = M_E(\tilde{o}_{ij}, \tilde{a}_{ij})$, where the $q_{ij}$ denotes that the utility is computed with respect to reference entity $j$, and the $M_E$ is the evaluation network in Fig. 2. We can select reference $j$ based on different strategies and obtain $q_i$ from $q_{ij}$ (which will be discussed in detail later). Once $q_i$ is obtained, the mixer network can compute $Q_{tot}$ as following:

$$Q_{\text{tot}} = w_1(s)q_1 + w_2(s)q_2 + \cdots + w_n(s)q_n + b(s), \tag{3}$$

where $w_i(s)$ are weights generated by a hypernetwork (conditioned on the global state $s$) that ensures all $w_i(s) \geq 0$ to maintain monotonicity. $b(s)$ is a state-dependent bias, also generated by the hypernetwork. In practice, the weights $w_i(s)$ and the bias $b(s)$ are produced by a separate hypernetwork that takes the state $s$ as input. This structure can factorize the joint action-value function $Q_{\text{tot}}$ while respecting the monotonicity condition, making decentralized execution possible.

During training, the loss for the final parameter update follows the Bellman equation as described in QMIX:

$$L(\eta) = \sum_{i=1}^{b} \left( r + \gamma \max \overline{Q}_{tot} - Q_{tot} \right)^2, \tag{4}$$

where the overline denotes the target network, the target network is typically updated with a delay, and its gradients are frozen during gradient ascent to enhance training stability (Mnih et al. (2013)).

**How to Select Reference Entity?** Regarding the selection of the reference entity $j$, we employ a simple yet feasible approach—alternately selecting each entity within the observation range as the reference target $j$. The final output (either $a_{ij}$ or $q_{ij}$) is averaged over all reference entities to reduce variance and enhance training stability. For example, the final action value $q_i = \frac{1}{n} \sum_{k=0}^{n} q_{ik}$, as this approach helps to reduce variance and improve training stability.

Further details on the algorithms and training procedures are provided in Appendix B.

## 5 EXPERIMENTS

In this section, we utilize experiments to validate the effectiveness of our method. Across two types of tasks, RDHNet demonstrates significant superiority when compared to state-of-the-art MARL algorithms for continuous action tasks.

### 5.1 TASK DOMAINS AND BASELINE

We tested the algorithm on two types of tasks. The first type is *Cooperative Prey Predator*, in which the prey moves at a slower speed following a fixed strategy, and the goal is to control the predtars to get as close to the prey as possible. The second type is a cooperative navigation task, where the objective is to control the agents to cover every target point. We adopted a simplified strategy in both tasks to focus on the research problem studied in this paper, where agents use global observation information in all algorithms. In both tasks, the agents' actions are force vectors, and there are obstacles present. The agents need to avoid obstacles as much as possible while achieving their task goals.

Through our investigation, we found that there are relatively few multi-agent reinforcement learning (MARL) methods specifically targeting continuous action spaces. Therefore, we selected FAC-MAC (Peng et al. (2021)), MADDPG (Lowe et al. (2017)), independent DDPG (Lillicrap et al. (2016)) (IDDPG), as well as COVDN (Peng et al. (2021)) and COMIX (Peng et al. (2021)), as baselines to evaluate the performance of our approach. COVDN and COMIX employ the cross-entropy method (CEM) (De Boer et al. (2005)) to successfully extend value decomposition-based methods, such as VDN Sunehag et al. (2018) and QMIX (Rashid et al. (2020)), to continuous action problems.

The following methods are all off-policy methods. We maintained consistency in the most common hyperparameters to ensure a fair comparison. Additionally, to mitigate the influence of randomness, all experiments in this study were conducted with different random seeds.

### 5.2 PERFORMANCE

#### 5.2.1 COMPARED WITH BASELINE

Fig. 4 illustrates the convergence performance of each algorithm during training. From the Fig. 4, we can observe that in the *Cooperative Navigation* tasks, RDHNet consistently converges quickly

Table 1: The Mean and Standard of Returns.

| Task | RDHNet (Ours) | COMIX | COVDN | FACMAC | MADDPG | IDDPG |
|---|---|---|---|---|---|---|
| cn(3 agent) | **-36.50±0.80** | -49.48±3.19 | -39.65±1.93 | -172.66±36.52 | -48.11±3.79 | -62.20±3.70 |
| cn(5 agent) | **-48.17±3.73** | -76.54±7.76 | -69.69±3.84 | -141.53±33.84 | -80.95±3.75 | -92.11±8365 |
| cn(7 agent) | **-61.91±4.05** | -97.41±9.47 | -88.91±2.36 | -211.83±115.19 | -97.43±4.28 | -117.51±15.39 |
| pp(3 predator) | 115.41±17.23 | 74.64±56.78 | **125.44±28.60** | 22.26±23.88 | 20.63±39.21 | 7.01±8.5 |
| pp(6 predator) | **280.80±16.35** | 15.39±9.26 | 65.82±55.98 | 38.86±45.96 | 8.43±4.96 | 13.03±4.01 |

and stably to the best results. Meanwhile, COMIX, COVDN, IDDPG, and MADDPG also achieve relatively good convergence, but FACMAC underperforms significantly, exhibiting poor convergence speed, final performance, and stability. In the *predator prey* tasks, RDHNet also achieves the fastest convergence and best performance across both scenarios. Furthermore, we observe that as the number of entities in the *predator prey* tasks increases, the advantages of RDHNet become more pronounced. Specifically, RDHNet demonstrates overwhelming superiority in the scenario with six predators compared to the other algorithms. We speculate that this is because the representation space is larger in scenarios with more entities, allowing RDHNet, which leverages rotational symmetry to compress the representation space, to have a more significant advantage. Tab. 1 lists the performance of all models after trained, showing that RDHNet achieves the best performance in four out of five tasks. The results confirm the objective presence of rotational symmetry in MARL tasks and demonstrate the effectiveness of our proposed method in addressing this issue.

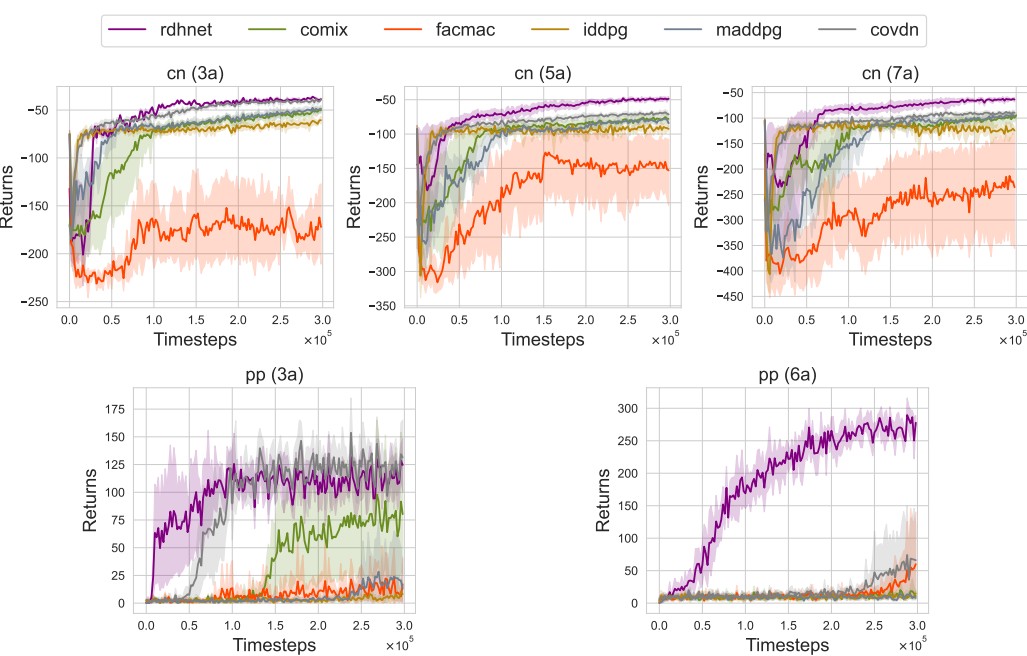

Figure 4: Performance comparison between RDHNet and other SOTA algorithms on five representative tasks.

## 5.3 ABLATION STUDY

To assess the impact of different symmetries in MARL, we conducted ablation experiments focusing on rotational and permutation symmetries. Given that rotational invariance is built upon permutation invariance, our experimental setup compared three scenarios: baseline, baseline with IE and PE, and baseline with only PE. The baseline used was COMIX, which is considered the best MARL algorithm for continuous action tasks. For implementing permutation invariance (PI), we replaced the MLP in COMIX with the HPN network. The method incorporating both permutation invariance (PI) and rotation invariance (IE) was our proposed RDHNet network.

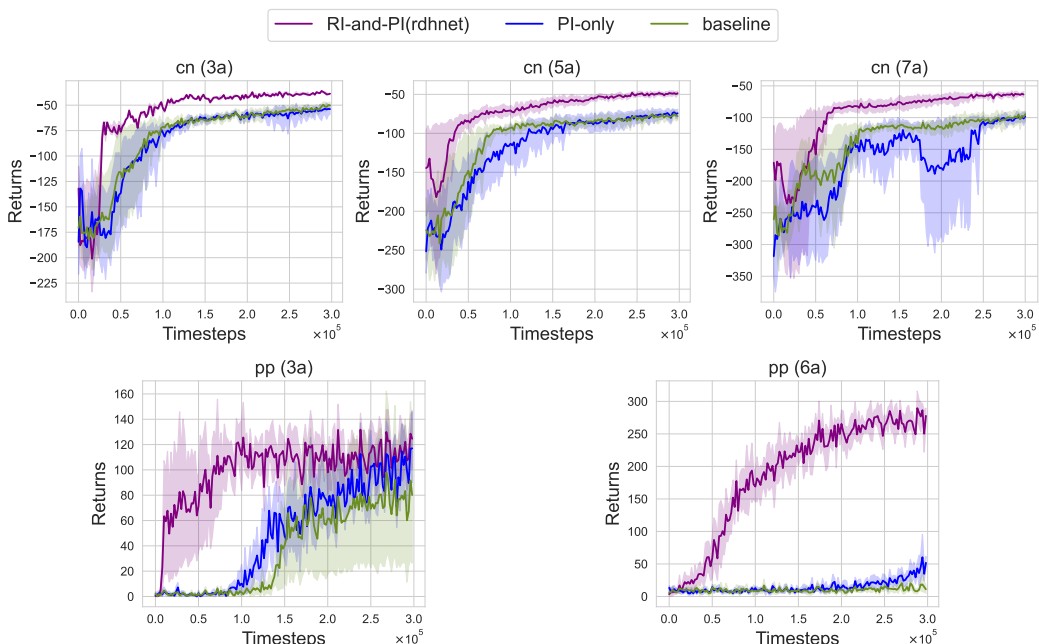

Figure 5: Ablation studies. The baseline is COMIX, PI-only is COMIX with HPN, and PI-and-RI is RDHNet.

The results, shown in Fig 5, reveal that introducing PE into the baseline improved performance across various Cooperative Prey Predator task scenarios compared to the original baseline. However, in the cooperative navigation task, performance did not significantly improve and even worsened as the number of entities increased. We speculate that the HPN network's use of PE reduced the policy's representational capacity. Nonetheless, the method incorporating both IE and PE achieved the best performance across all task scenarios, with its advantage becoming more pronounced as the number of entities increased. These experiments confirm that symmetry indeed exists in MARL problems and has a tangible impact on MARL performance.

## 6 CONLUSION AND DISCUSSION

In this paper, we first thoroughly analyze the representation redundancy problem caused by symmetry in multi-agent scenarios, distinguishing between permutation symmetry and rotational symmetry. We then propose the RDHNet architecture, inspired by human observation and problem-solving approaches, to address the issue of rotational symmetry in multi-agent systems. Finally, we validate our approach through experiments that reveal the objective hindrance posed by rotational symmetry to multi-agent learning and demonstrate that RDHNet applies to a wide range of multi-agent scenarios, further supporting our hypothesis: in specific scenarios and environments, Cartesian coordinates are unnecessary, as human intuition perceives distance and angles rather than constructing a Cartesian coordinate system in the mind.

We also acknowledge certain limitations of this work. For instance, the experimental scenarios lack diversity, the algorithm's computational complexity increases quadratically with the number of entities in the system, and we have yet to develop more effective optimization strategies for the actor-critic version of RDHNet. These aspects will be the focus of our future work.

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

## A  MATHMATICAL PROOF

Assume that at a certain moment, the observation of agent $i$ is:

$$o_i = [E_1, \ldots, E_n] = [(p_1, z_1), \ldots, (p_n, z_n)] = [((x_1, y_1), z_1), \ldots, ((x_n, y_n), z_n)],$$

where $x_k, y_k$ denotes the Cartesian coordinates corresponding to entity $k$. Next, I can derive the formula for calculating the angle information as follows:

$$\sin(\theta_{ij}) = \frac{y_j}{\sqrt{x_j^2 + y_j^2}}, \quad \cos(\theta_{ij}) = \frac{x_j}{\sqrt{x_j^2 + y_j^2}},$$

$$\sin(\theta_{ik}) = \frac{y_k}{\sqrt{x_k^2 + y_k^2}}, \quad \cos(\theta_{ik}) = \frac{x_k}{\sqrt{x_k^2 + y_k^2}},$$

$$\sin(\theta_{ijk}) = \sin(\theta_{ik} - \theta_{ij}) = \sin(\theta_{ik}) \cdot \cos(\theta_{ij}) - \cos(\theta_{ik}) \cdot \sin(\theta_{ij}),$$

$$\cos(\theta_{ijk}) = \cos(\theta_{ik} - \theta_{ij}) = \cos(\theta_{ik}) \cdot \cos(\theta_{ij}) + \sin(\theta_{ik}) \cdot \sin(\theta_{ij}),$$

$$m_{ijk} = [\sin(\theta_{ijk}), \cos(\theta_{ijk})]$$

$$= \left[ \frac{y_k x_j}{\sqrt{(y_k^2 + y_k^2)(x_j^2 + y_j^2)}} - \frac{x_k y_j}{\sqrt{(x_k^2 + y_k^2)(y_j^2 + y_j^2)}}, \right.$$

$$\left. \frac{x_k x_j}{\sqrt{(x_k^2 + y_k^2)(x_j^2 + y_j^2)}} + \frac{y_j y_k}{\sqrt{(y_k^2 + y_k^2)(y_j^2 + y_j^2)}} \right].$$

In the above derivation, $\theta_{ij}$ represents the angle between the line connecting entity $i$ and entity $j$ and the x-axis of the Cartesian coordinate system. Meanwhile, $\theta_{ijk}$ denotes the angle formed by entities $i$, $j$, and $k$, where $i$ is the vertex of the angle. Other notations follow a similar representation.

## B IMPLEMENTATION DETAILS OF COMIX-BASED RDHNET AND EXPERIMENTAL HYPERPARAMETER SETTING

The experimental results related to RDHNet in this paper are implemented based on COMIX, and the detailed processing steps of RDHNet are outlined in Algorithm 1. At the same time, Algorithm 2 provides a detailed description of the process by which RDHNet, combined with CEM (De Boer et al. (2005)), outputs actions and their utility.

---

**Algorithm 1:** RDHNet based on COMIX

---

**input** : $\xi$: RDHNet
  $\omega$:Mixer network
  $T$: total timestep
  $n$: number of agents
  $K_i$: timesteps of the parameters update interval
  $\mathcal{B}$: replay buffer
  $K_b$: sampling threshold of the replay buffer

1 *Initialize all network parameters*;
2 *Initialize the replay buffer $\mathcal{B}$*;
3 **for** $t = 1, 2, ..., T$ **do**
4   *Obtain each agent's observation $\mathbf{o} = \{o_i\}_{i=1}^n$ and global state $s$*;
5   **for** $agent\, i = 1, 2, ..., n$ **do**
6     **foreach** $agent \in \mathcal{N}(i)$ **do** *compute $q_i$ and $a_i$ by **Algorithm** 2 with inputing $(o_i^t, \xi)$* ;
7   *Execute joint action $\mathbf{a} = \{a_i\}_{i=1}^n$, and obtain reward $r$*;
8   *Store $(s, \mathbf{o}, \mathbf{a}, r)$ to $\mathcal{B}$*;
9   **if** $t \bmod T_i = 0$ and $|\mathcal{B}| > K_b$ **then**
10     *Sample a batch of trajectories from $\mathcal{B}$*;
11     *Compute $Q_{tot} = f_\omega([q_1, q_2, ..., q_n])$ for each data by Equation 3*;
12     *Compute the batch data loss $\sum L(\xi, \omega)$ by Equation 4*;
13     *Update parameters $\xi, \omega$ by gradient descent*;

**output:** RDHNet with the optimal parameters $\xi^*$

---

---

**Algorithm 2:** CEM with RDHNet

---

**input** : $\xi$: RDHNet
  $o_i$: observation of agent $i$
  $K_c$: iteration number of CEM
  $K_s$: number of samples in every CEM iteration

1 *Initialize the Gaussian distribution $U(\mu, \sigma^2)$ as a standard normal distribution*;
2 **for** $c = 1, ..., k_c$ **do**
3   *Sample $k_s$ actions $\{a_i^k\}_{k=1}^{K_s}$ from $U(\mu, \sigma^2)$*;
4   **foreach** $agent\, j \in \mathcal{N}(i)$ **do**
5   *Compute: $q_{ij}^k = f_\phi(o_i, a_i^k, p_j)$ for each elements in $\{a_i^k\}_{k=1}^{K_s}$ by Equation (1~ 2)*;
6   *Compute the optimal action $a_i^* = \arg\max_k q_{ij}^k$*;
7   *Compute the utility of the optimal action $q_{ij} = \max_k(q_{ij}^k)$*;
8   *Let $\mu = a_i^*$*;
9 *Compute the utility of agent $i$: $q_i = \frac{1}{|\mathcal{N}(i)|} \sum_j q_{ij}$*;
10 *Let $a_i = a_i^*$*.
**output:** $q_i, a_i$

---

The setting of mixer network is kept the same as that of QMIX (Rashid et al. (2020)), and the details of the network is shown in Tab. 2

Table 2: The network configurations used for RDHNet.

| Network Configurations | Value | Network Congfigurations | Value |
|---|---|---|---|
| embedding mlp dim | 64 | rbf type | gaussian |
| embedding mlp layer | 2 | rbf lower bound | 0 |
| hypernet head | 4 | rbf upper bound | 10 |
| hypernet output dim | 9 | output fc dim | 64 |
| hypernet hidden dim | 64 | output fc layer | 2 |

To ensure the fairness of the comparative experiments, we made every effort to keep the common parameters consistent. Since all the methods used in this paper are off-policy, maintaining parameter consistency is relatively easy. The parameters listed in Tab. 3 are common to all methods, and thus they remain the same across all experiments. This strict control of parameter consistency is crucial to ensure the validity of our experimental results.

Table 3: Hyperparameters used for RDHNet based on COMIX.

| Hyperparameter | Value | Hyperparameter | Value |
|---|---|---|---|
| buffer size | 5000 episodes | act noise | 0.1 |
| batch size | 32 | evaluate interval | 2000 |
| learning rate | 0.01 | grad norm clip | 0.5 |
| exploration mode | gaussian noise | optimizer | adam |
| discount factor | 0.99 | target update interval | 200 |

## C    LATENT EMBEDDING VISUALIZATION

In this section, we use the data distribution in the latent space to demonstrate RDHNet's capability to compress the representation space effectively. Specifically, we first interact with the environment using a random policy to collect a dataset $\mathcal{D}$. Next, we apply rotation to the data, rotating each sample by arbitrary angles (in practice, each data point generates ten rotated versions). After constructing the dataset $\mathcal{D}$, each $(o, a)$ pair in the dataset is input into different models. We then extract the features from the layer preceding the output layer as the data representation in the latent space described by the model.

These representations are clustered using t-SNE, with the results shown in Fig. 6. The figure shows that the representations of the baseline (COMIX) are the most dispersed in the latent space, followed by the baseline augmented with Permutation Invariance (PI). The most compact clustering is produced by the model that incorporates both Permutation Invariance and Rotational Invariance (RI), which is RDHNet. In our experimental setup, the degree of clustering reflects the compactness of the representation space. Thus, we have sufficient reason to believe that integrating both PI and RI in our RDHNet model enables the effective compression of redundant representations, as demonstrated in Fig 6.

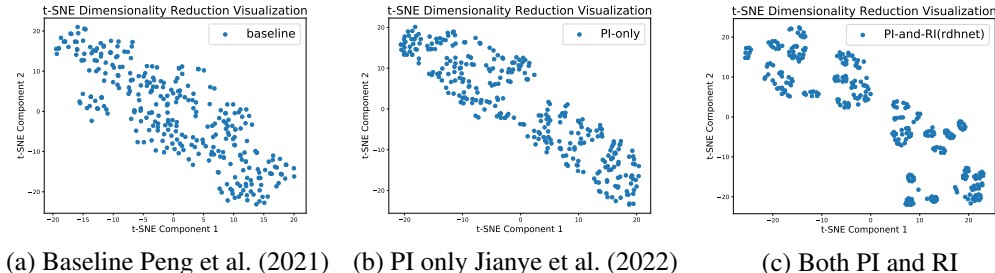

(a) Baseline Peng et al. (2021)    (b) PI only Jianye et al. (2022)    (c) Both PI and RI

Figure 6: The t-SNE visualization of features with different methods in latent space.

