# OpenReview forum: "RDHNet: Addressing Rotational and Permutational Symmetries in Continuous Multi-Agent Systems"
_ICLR.cc/2025/Conference — ICLR 2025 Conference Withdrawn Submission_

### Official Review · Reviewer_LRjA · 2024-10-20

**Soundness:** 2
**Presentation:** 3
**Contribution:** 2
**Rating:** 5
**Confidence:** 3

**Summary:**

The paper presents RDHNet, a novel approach for addressing rotational symmetries in multi-agent reinforcement learning (MARL) systems with continuous action spaces. Rotational symmetry in MARL introduces redundant state representations, which can hinder learning efficiency. RDHNet introduces a rotation-invariant architecture that utilizes relative coordinate systems and hypernetworks to enhance its ability to model complex multi-agent dynamics.

**Strengths:**

The authors formalize the symmetry problem in MARL and distinguishing between permutation and rotational symmetry. They propose a novel RDHNet architecture, which extracts relative directional and positional information, compressing redundant representations caused by symmetry. The empirical results demonstrate the superiority of the proposed method over baselines.

**Weaknesses:**

1. The authors considers coordinate transformation to deal with the redundancy problem. Would this coordinate transformation misunderstand the meaning of the original observation and further affects the action-decision making.
2. The authors should give more explanations on why coordinate transformation can reduce redundancy.
2. When the number of agents in the environment changes, can the original network structure still be applied to this change.
3. Whether the increased network complexity would affect the learning efficiency.

**Questions:**

See the weaknesses above.

---

### Official Review · Reviewer_9sp5 · 2024-10-22

**Soundness:** 2
**Presentation:** 2
**Contribution:** 1
**Rating:** 3
**Confidence:** 4

**Summary:**

This paper presents RDHNet to address rotational and permutational symmetry in comtinuous MARL. The author propose a rotation-
invariant network for continuous action space, which utilize relative coordinate between agents, and use a hypernet to enehance the fitting capability of models. Experiments in cooperative navigation and predator prey demonstrates the effectiveness of the proposed algorithm.

**Strengths:**

The paper deals with an important problem of aggregating real-world rules in MARL algorithms. The writing is easy to follow.

**Weaknesses:**

1. The main contribution proposed by authors is a method to handle continuous transformations. However, this seems only a minor technical detail in achieving symmetry. Also, while authors claim "They neither consider nor can be applied to continuous random rotational symmetry, which is precisely the focus of our work and is more aligned with real-world scenarios", invariance to contiuous transformations are already studied in [1, 2]. So I wonder what are the contributions made by authors.

2. The proposed method lack theoretical guarantees and seems largely empirical. I would recommend authors to add additional formal analysis for the proposed method.

3. MARL should be considered as a Markov Game or Dec-POMDP, not a MDP, as stated in Section 3. The problem stated by author sees more like a Dec-POMDP, which is cooperative MARL. This should be explicitly stated.

4. The authors could consider evaluating their method on some real-world tasks instead of toy simulations to better demonstrate their applicability in "real-world scenarios".

Minors: please check the typos, such as Sec. 4.3, ALGORITHM INPLEMENTATION should be ALGORITHM IMPLEMENTATION. Also check grammar errors.

[1] Equivariant Actor-Critic Methods for Cooperative Multi-Agent Reinforcement Learning. ICML 2024.

[2] Boosting Sample Efficiency and Generalization in Multi-agent Reinforcement Learning via Equivariance. Arxiv 2024.

**Questions:**

Please see "weakness" section.

---

### Official Review · Reviewer_fDjv · 2024-10-26

**Soundness:** 3
**Presentation:** 2
**Contribution:** 1
**Rating:** 1
**Confidence:** 3

**Summary:**

The authors propose a network architecture for multiagent RL problems where absolute coordinates are autonomously converted to rotation invariant features.

**Strengths:**

- On the very high level, the authors investigate an important problem: bisimulation, or how to compute similarity between different states to find representations where equivalent states are merged.

**Weaknesses:**

- Pretty much the whole of the paper is only applicable to domains in which the state is described through coordinates, which not only is a very restricting assumption, but also if this is the case of the application of interest, it sounds to me trivial to just change the state representation to use rotation-invariant coordinates, instead of having a dedicated layer to perform this translation.

- I suggest the authors focus instead in developing an architecture able to identify autonomously equivalent states (that is not only applicable in navigation domains).

- A much more complex experimentation evaluation will also be needed, as well as the incorporation of benchmarks of other approaches that compute state similarity.

**Questions:**

No specific question.

---

### Note · Authors · 2024-11-18

I have read and agree with the venue's withdrawal policy on behalf of myself and my co-authors.